# Identification of Clonality through Genomic Profile Analysis in Multiple Lung Cancers

**DOI:** 10.3390/jcm9020573

**Published:** 2020-02-20

**Authors:** Rumi Higuchi, Takahiro Nakagomi, Taichiro Goto, Yosuke Hirotsu, Daichi Shikata, Yujiro Yokoyama, Sotaro Otake, Kenji Amemiya, Toshio Oyama, Hitoshi Mochizuki, Masao Omata

**Affiliations:** 1Lung Cancer and Respiratory Disease Center, Yamanashi Central Hospital, Yamanashi 400-8506, Japan; lumi.hgc.236@gmail.com (R.H.); nakagomi.takahiro@gmail.com (T.N.); shikarupd@yahoo.co.jp (D.S.); dooogooodooo@me.com (Y.Y.); sotaro.otake@gmail.com (S.O.); 2Department of Surgery, School of Medicine, Keio University, Tokyo 160-8582, Japan; 3Genome Analysis Center, Yamanashi Central Hospital, Yamanashi 400-8506, Japan; hirotsu-bdyu@ych.pref.yamanashi.jp (Y.H.); amemiya-bdcd@ych.pref.yamanashi.jp (K.A.); h-mochiduki2a@ych.pref.yamanashi.jp (H.M.); m-omata0901@ych.pref.yamanashi.jp (M.O.); 4Department of Pathology, Yamanashi Central Hospital, Yamanashi 400-8506, Japan; t-oyama@ych.pref.yamanashi.jp; 5Department of Gastroenterology, The University of Tokyo Hospital, Tokyo 113-8655, Japan

**Keywords:** lung cancer, multiple cancers, metastasis, sequencing, mutation, genomic diagnosis

## Abstract

In cases of multiple lung cancers, individual tumors may represent either a primary lung cancer or both primary and metastatic lung cancers. In this study, we investigated the differences between clinical/histopathological and genomic diagnoses to determine whether they are primary or metastatic. 37 patients with multiple lung cancers were enrolled in this study. Tumor cells were selected from tissue samples using laser capture microdissection. DNA was extracted from those cells and subjected to targeted deep sequencing. In multicentric primary lung cancers, the driver mutation profile was mutually exclusive among the individual tumors, while it was consistent between metastasized tumors and the primary lesion. In 11 patients (29.7%), discrepancies were observed between genomic and clinical/histopathological diagnoses. For the lymph node metastatic lesions, the mutation profile was consistent with only one of the two primary lesions. In three of five cases with lymph node metastases, the lymph node metastatic route detected by genomic diagnosis differed from the clinical and/or pathological diagnoses. In conclusion, in patients with multiple primary lung cancers, cancer-specific mutations can serve as clonal markers, affording a more accurate understanding of the pathology of multiple lung cancers and their lymphatic metastases and thus improving both the treatment selection and outcome.

## 1. Introduction

In patients with synchronous or metachronous multiple cancers, individual tumors may appear as either a primary lung cancer or both primary and metastatic lung cancers. The selection of treatment in such cases is dependent on the resulting characteristics. In patients with multiple lung cancers, the nature of a tumor (i.e., whether it is metastatic or primary) can usually be judged on the basis of diagnostic imaging findings, clinical course, and/or pathology. If individual tumors composing multiple lung cancers are histologically inconsistent in terms of histological morphology and/or cellular atypism, the multiple onset of primary cancers is highly likely. However, there are no specific radiological, clinical or histological features that can be utilized to unambiguously distinguish intrapulmonary metastases from multiple primary cancers and the cut diagnosis can be perplexing in the clinical setting. The differing biological activities of tumors allow for prognostic distinctions to be drawn and patients with intrapulmonary metastasis are supposed to have a poorer prognosis. Therefore, it is critically important to develop improved methods for the identification of tumors by exploring new, practical techniques and markers. We have previously demonstrated that as a more precise and clinically applicable method, a comparison of the driver mutation profiles enables elucidation of the clonal origin of tumors and thus facilitates an accurate discrimination between primary and metastatic tumors [1]. However, this finding was based on only 12 multiple lung cancer cases; hence, validation through a study involving a larger number of such cases was needed. Moreover, the significance of these findings in the clinical setting remained to be determined. In view of this, we extended the case accrual period to 5 years and included 37 patients with multiple lung cancers in the present study. In addition, we analyzed the clinical course in individual patients in detail to examine the use of mutation data for the diagnosis of multiple lung cancers in clinical practice and to determine the actual contribution of this approach to an improvement of clinical practice. Furthermore, we analyzed gene mutations in primary lung cancers as well as metastatic lymph nodes and genetically examined the pathology of the metastatic lymph nodes to accurately understand the pathology of lymphatic metastasis and thus enhance the postoperative treatment outcome.

## 2. Methods

### 2.1. Patients and Sample Preparation

The study enrolled 37 patients who had undergone surgery for multiple lung cancers in our department between January 2015 and July 2019. Written informed consent for genetic research was obtained from all patients, which was performed in accordance with protocols approved by the institutional review board in our hospital. Histological typing was performed according to the World Health Organisation (WHO) classification (3rd edition) [2] and clinical staging was performed according to the International Union Against Cancer Tumor-Node-Metastasis (TNM) classification (8th edition) [3].

A serial section from formalin-fixed, paraffin-embedded (FFPE) tissue was stained with hematoxylin-eosin and subsequently microdissected using an ArcturusXT laser capture microdissection system (Thermo Fisher Scientific, Tokyo, Japan). DNA was extracted using the QIAamp DNA FFPE Tissue Kit (Qiagen, Tokyo, Japan). FFPE DNA quality was verified using primers for the ribonuclease P locus. Peripheral blood was drawn from each patient immediately before surgery. A buffy coat was isolated by centrifugation and DNA was extracted from these cells using the QIAamp DNA Blood Mini Kit (Qiagen).

### 2.2. Targeted Deep Sequencing and Data Analysis

A panel covering the exons of 53 lung cancer-related genes (see Appendix A) was designed in-house to perform targeted sequencing. These genes were selected after a literature search based on the following criteria: (a) genes involved in lung cancer according to The Cancer Genome Atlas [4,5] and other, similar projects [6,7,8,9,10] or (b) genes frequently mutated in lung cancer according to the Catalogue Of Somatic Mutations In Cancer (COSMIC) database [11]. Ion AmpliSeq designer software (Thermo Fisher Scientific) was utilized for the primer composition, as previously reported [1,12,13]. An Ion AmpliSeq Library kit (Thermo Fisher Scientific) was utilized for the preparation of sequencing libraries. The library samples were bar-coded with an Ion Xpress Barcode Adapters kit (Thermo Fisher Scientific), purified using Agencourt AMPure XP reagent (Beckman Coulter, Tokyo, Japan) and subsequently quantified using an Ion Library Quantitation Kit (Thermo Fisher Scientific). The libraries were templated with an Ion PI Template OT2 200 Kit v3 (Thermo Fisher Scientific). Sequencing was performed on Ion Proton (Ion Torrent) with an Ion PI Sequencing 200 Kit v3.

The sequence data were processed on standard Ion Torrent Suite Software. Raw signal data were measured using the Torrent Suite version 4.0. The pipeline consisted of signaling processing, base calling, quality score assignment, read alignment to the human genome 19 reference (hg19), mapping quality control and coverage analysis. After the data analysis, the annotation of single-nucleotide variants and indels (insertions and deletions) was performed on the Ion Reporter Server System (Thermo Fisher Scientific). Blood cell DNA extracted from the peripheral blood was used as a normal control to detect variants (Tumor-Normal pair analysis). Sequencing data were visually analyzed using an Integrative Genomics Viewer.

## 3. Results

### 3.1. Patient Characteristics

The 37 patients recruited in this study (age range, 54–85 years; mean age, 70.5 ± 7.5 years) were divided into different groups according to the following characteristics (Appendix A): 31 males, 6 females; 30 smokers, 7 non-smokers; and pathological stage IA (n = 10), IB (n = 15), IIA (n = 2), IIB (n = 4), IIIA (n = 5) and IIIB (n = 1). The maximum tumor diameter ranged from 2 mm to 80 mm (mean tumor diameter, 24.5 ± 15.9 mm).

Twenty nine patients were diagnosed with double or triple primary lung cancers on the basis of histopathological characteristics, including 15 patients with adenocarcinoma–adenocarcinoma, 3 patients with squamous cell carcinoma–squamous cell carcinoma, 5 patients with adenocarcinoma–squamous cell carcinoma and 6 patients with other combinations. In terms of tumor development, tumors developed synchronously and metachronously in 26 and 11 patients, respectively. In patients with metachronous tumors, the tumors were designated as tumor 1 (T1), T2 and T3 in chronological order from the earliest to the latest. In those with synchronous tumors, this designation was based on the order of size from the largest to the smallest.

### 3.2. Targeted Sequencing Identified Somatic Mutations in the Lung Cancers

Targeted sequencing was performed on 76 surgically resected tumors and 8 lymph nodes obtained from 37 patients, with their blood cell samples utilized as normal controls. The mean coverage depth was 1411-fold for cancer samples (range, 106- to 5096-fold) and 1387-fold for blood cell samples (range, 76- to 6960-fold). Sequence analyses detected 314 somatic mutations with an allele fraction ≥1% from 84 cancer lesions (1–54 mutations per tumor) (Appendix A). Among these mutations, 137 mutations (44%) were present at an allele fraction ≥20% (Appendix A).

In 29 patients, the gene, amino-acid substitution and nucleotide changes that were caused by these somatic mutations within individual tumors composing the multiple lung cancers lacked consistency (Figure 1, Appendix A). Thus, there were no shared or overlapping mutations among the individual lung cancers detected in these patients. This finding demonstrated that the multiple lung cancers in these cases were independently developed primary lung cancers (Figure 1). Meanwhile, in 8 patients, the gene mutation profile was consistent among the individual tumors, suggesting the presence of intrapulmonary metastasis (Figure 2). Importantly, in these cases, nucleotide position and mutation variance were entirely consistent across the tumors (Appendix A).

### 3.3. Case Presentations

#### Three Representative Cases are Described in Detail Below

##### Case A (Case 30 in Table 1 and Appendix A)

A 74-year-old man had two tumors in the right upper lobe that were resected through right upper lobectomy. Both tumors morphologically had an irregular surface; thus, they were diagnosed as primary lung cancers (Figure 3A,B). Pathologically, the peripheral lesion was identified as an adenosquamous carcinoma comprised of squamous cell carcinoma and acinar-predominant adenocarcinoma, whereas the central lesion was identified as papillary-predominant adenocarcinoma (Figure 3C,D). On the basis of the histopathological differences, the tumors were judged as double primary tumors. Pathologically, the cancer stage was determined to be pT1cN2M0, stage IIIA. However, the genetic mutation profiles were completely consistent between these two tumors, suggesting they are metastases (Figure 3E). Moreover, their mutation profiles were also consistent with the mutation profile of the metastatic lymph node. (Figure 3E). Based on the genetic diagnosis, the cancer stage was ultimately upgraded to T3N2M0, stage IIIB. At the patient’s request, he was placed on follow-up without any postoperative adjuvant chemotherapy. The patient has remained alive for 2 years postoperatively without any recurrence.

##### Case B (Case 10 in Table 1 and Appendix A)

A 59-year-old woman presented with 0.7-cm nodules in the right lower lobe 1.5 years after undergoing right upper lobectomy for cancer. The tumors were round and had a smooth surface. Because of their morphology, they were suspected of being metastatic lesions. After 4 months of follow-up, there was no increase in the number of lung lesions, suggesting solitary intrapulmonary metastasis. Subsequently, wedge resection was performed. Although both tumors were pathologically papillary-predominant adenocarcinoma (Figure 4C,D), a lepidic pattern was observed in the periphery of the smaller nodule (Figure 4E), leading to a diagnosis of double primary lung cancers. However, the genetic mutation profile was consistent between the two tumors, suggesting them to be metastases (Figure 4F). The patient was positive for a mutation in the epidermal growth factor receptor (EGFR) gene (exon 19 deletion); hence, oral administration of an EGFR-tyrosine kinase inhibitor (gefitinib) was continued. The patient has remained alive without recurrence for 4 years after the second surgery.

##### Case C (Case 18 in Table 1 and Appendix A)

A 74-year-old man presented with tumors measuring 4.0 cm and 1.8 cm in the left upper lobe, so left upper lobectomy was performed (Figure 5A,B). As both tumors were closely located and pathologically similar squamous cell carcinomas, they were assumed to be single origin pulmonary metastases (Figure 5C,D). However, the mutation profile was completely different between the two tumors genetically, suggesting double primary cancers (Figure 5E).

### 3.4. Investigation of the Discrepancies between the Clinical and/or Histopathological Diagnoses and Genetic Diagnosis

Table 1 shows the discrepancies between and among the clinical, pathological and genetic diagnoses of the primary or metastatic lesions in all 37 patients. The clinical diagnoses were comprehensively determined, mainly on the basis of imaging findings and clinical course by the cancer board of the hospital (comprised of thoracic surgeons, pulmonologists, pathologists and radiologists). The pathological diagnoses were determined on the basis of the postoperative pathological findings, especially the differences in the tissue morphology and cellular atypia detected by pathologists. The genetic diagnoses were determined on the basis of digital and statistical analyses of overlaps in the mutation profiles of individual tumors. Discrepancies between the genetic diagnosis and clinical and/or histopathological diagnoses were observed in 11 patients (29.7%). In the patients with synchronous tumors, primary and metastatic tumors were eventually diagnosed on the basis of genetic diagnosis in 24 and 2 patients, respectively. In the 11 patients with metachronous tumors, primary and metastatic tumors were diagnosed in 5 and 6 patients, respectively, in the same manner. The distribution of primary and metastatic tumors between synchronous and metachronous tumors was significantly different; thus synchronous multiple lung tumors were deemed likely to be primary lesions.

### 3.5. Genetic Diagnosis of Lymph Node Metastasis in Patients with Multiple Lung Cancers

Lymph node metastasis was detected in five patients with double primary lung cancers (Table 2). It occurred approximately at the time of surgery in three patients and was identified as postoperative lymph node recurrence in two patients (Table 2). In some patients, the route of lymph node metastasis was apparent from the timing of the metastasis as well as the location and pathological findings of the metastatic lesions (cases 4 and 5 in Table 2). In contrast, it was difficult to identify the clonal origin of lymph node metastasis on the basis of the clinical and pathological findings in the other patients, especially in those in whom the primary lesions were both squamous cell carcinomas (cases 1–3 in Table 2). However, even in these patients, a comparison of the mutation profiles of the primary and lymph node metastatic lesions revealed the route of lymph node metastasis (Figure 6).

### 3.6. Case Presentations

#### Three Representative Cases are Described in Detail Below

##### Case D (Case I in Table 2 and Figure 6)

A 74-year-old man, described as case C in the previous section, presented with paratracheal and mediastinal lymph node metastases 1 year after left upper lobectomy (Figure 5F). Although it was not possible to pathologically identify the metastasizing primary lesion (Figure 5G), the mutation profile of the metastatic lymph node was genetically consistent with that of the larger cancer. The genetic diagnosis was lymph node metastasis of the larger cancer (Figure 6).

##### Case E (Case II in Table 2 and Figure 6)

A 77-year-old man with lung cancer underwent left lower lobectomy (Figure 7A). One year later, a nodule appeared in the middle lobe (Figure 7B). Middle lobectomy was performed based on the assumption that the lesion was a double primary tumor. However, after 1 year, subcarinal lymph node metastasis occurred (Figure 7C). Pathologically, all three lesions were of squamous cell carcinoma type and it was impossible to determine which primary lesion had metastasized (Figure 7D–F). Given the tumor size, the tumor in the left lobe was clinically more likely to have metastasized. However, mutation analysis revealed that the two lung lesions had different mutation profiles; therefore, they were diagnosed as double primary lung cancers. Furthermore, the mutation profiles were consistent between the middle lobe lung cancer and the metastatic lymph node. Thus, lymph node metastasis of the middle lobe lung cancer was determined (Figure 6). Programmed death-ligand 1 (PD-L1) staining of tumor cells was 0% and 90% in the left lower lobe and middle lobe tumors, respectively. Treatment with an anti-PD-1 antibody (nivolumab) was administered and a complete response has been maintained for 1 year since the recurrence in the lymph node.

##### Case F (Case III in Table 2 and Figure 6)

A 72-year-old man presented with two tumors in the right lower lobe. Imaging findings suggested double primary lung cancers and right lower lobectomy was performed (Figure 7G,H). Postoperative pathological examination revealed metastases in the interlobar and subcarinal lymph nodes. All four lesions, including the double primary lesions and two metastatic lymph nodes, were pathologically similar squamous cell carcinomas. Therefore, it was impossible to determine which primary lesion had metastasized to the lymph nodes (Figure 7I–L). Clinically, the larger segment 9 tumor was likely to have metastasized to the two lymph nodes. However, both segment 6 and 9 tumors, which had different mutation profiles, were genetically identified as double primary lung cancers. In addition, it was found that the larger segment 9 tumor had metastasized to the interlobar lymph node, whereas the smaller segment 6 tumor had metastasized to the subcarinal lymph node (Figure 6). PD-L1 staining of tumor cells was 0% and 70% in the segment 9 and segment 6 tumors, respectively. Despite the administration of an anti-PD-1 antibody (nivolumab), the patient did not respond to the treatment and died of progression of the cancer at 17 months postoperatively.

## 4. Discussion

In cases of multiple lung cancers, clinical differentiation between primary and metastatic tumors can be difficult, rendering treatment selection challenging. Furthermore, in patients with multiple lung cancers metastasized to the lymph nodes or distal sites, the focus of treatment varies depending on the cancer that has metastasized. Thus, determining the origin of the metastasizing cancer is clinically important. Therefore, we performed lung cancer mutation analysis through targeted deep sequencing and demonstrated that mutations of individual lung cancers are able to provide clonal markers, enabling discrimination of the clonal origin of multiple lung cancers and their metastases.

The consistency of mutations across multiple sites, with complete concordance in the position and patterns of base-pair substitutions or indels, cannot be a coincidental phenomenon. Although discordance between two tumors was noted in mutations with an allele fraction <20%, this can be interpreted as tumor heterogeneity [14]. In general, cancers comprise populations of cells with various molecular and phenotypic features, a phenomenon termed intratumor heterogeneity [14,15]. This may bolster tumor adaptation, cancer progression and metastasis, and/or therapeutic failure through negative selection [14,16]. Conversely, a driver mutation triggers clonal expansion and is retained ubiquitously within the tumors of the same clone [16,17]. These theories can be interpreted as the “trunk and branch” mutation models; early somatic events that drive tumor progress in early clonal founders are represented by the “trunk” of the tumor [18,19]. Such trunk somatic mutations to be found at the early stages of tumor development are ubiquitous events occurring at all sites of disease. Meanwhile, later somatic events that occur in the wake of branched separation of subclones represent heterogeneity. Such subclonal heterogeneity may be spatially divided among regions of the same tumor or its metastatic sites [18,19,20]. In this context, clonally dominant mutations are important clonal markers. Primary and metastatic tumors can be differentiated by determining whether such ubiquitous driver mutations are consistent.

It is relatively straightforward to diagnose multicentric primary lung cancers of different histological type. However, it is often difficult to differentiate between multiple primary lung cancers and intrapulmonary metastases having the same histological type. In particular, in cases of multiple tumors classified as squamous cell carcinoma (such as cases C–F), differentiation based on pathological features alone is extremely difficult. Even when the morphological and immunohistological features are non-homogeneous among different parts of the tumors (e.g., cases A and B), the driver mutation is ubiquitously retained within the tumors of the same clone [16,17]. Therefore, distinction of clonality on the basis of mutation analysis is more specific and definitive than histological examination.

Detterbeck et al. reviewed the clinical and pathological criteria to distinguish second primary tumors from metastatic tumors [21]. They reported that it is impracticable to define criteria that conclusively establish the identical nature of tumors; merely finding observable similarities between tumors is insufficient. Using the method described, comprehensive mutation analysis is initially performed to identify the driver mutations in each cancer, which are subsequently compared to define their clonal origin. These criteria are definitive and reliable. Moreover, the decision criteria are generally clear and intuitive. In fact, this method yielded clear genetic diagnosis in all patients. In other words, no equivocal or ambiguous diagnosis was obtained in any of the cases. In our previous study, we had demonstrated that this method allows bronchoscopic biopsy samples and other small samples to be used for discrimination between primary and metastatic tumors [1]. Thus, our method may enable both flexible and rational decision-making based on accurate diagnosis. For example, a preoperative diagnosis of metastatic tumors may make it possible to avoid surgery, whereas a preoperative diagnosis of primary tumors would lead to surgical treatment. This novel approach may help resolve the dilemma of misdiagnosis in the clinical setting. Thus, we anticipate that it will come to be utilized as a standard diagnostic approach in daily clinical practice in the near future.

When selecting treatment methods for multiple lung cancers, it is necessary to consider the cancer type will markedly affect the prognosis. In cases D–F that have lymph node metastasis, a factor responsible for progression to an advanced stage was identified in the two tumors. Furthermore, the tumors exhibit different mutation profiles and PD-L1 staining properties. Therefore, the lesions targeted for treatment and the options selected for subsequent treatment (e.g., molecular-targeted drugs and immune checkpoint inhibitors) vary depending on the type of tumor that has metastasized to the lymph nodes. This suggests that accurate understanding of the pathology gained by performing a genetic diagnosis can exert a powerful effect on the clinical outcome. Although the use of immunotherapy has revolutionized the treatment of non-small-cell lung cancer, patterns of immunostaining with PD-L1, a biomarker for treatment response, may vary in tumor cells across individual primary tumors (e.g., (,) cases E and F) [22]. At present, molecularly targeted therapies are also rapidly evolving. The development of novel molecularly targeted therapies would enable the treatment to be specifically tailored to the features of mutations detected in individual cancers. Thus, in patients with multiple lung cancers, performing a mutation analysis helps select the medical treatment most likely to be effective.

## 5. Conclusions

In cases of multiple lung cancers, identifying the differences in the mutation profiles of multiple tumors will help determine their clonal origin and enable a distinction to be drawn between primary and metastatic tumors with great specificity, even in cases in which pathological distinction is impossible or equivocal. In addition, performing genetic diagnosis in addition to pathological diagnosis can help obtain a more accurate understanding of the pathology of multiple lung cancers and the lymphatic metastases. This approach may lead to the provision of treatment specifically tailored to the features of individual cases.

## Figures and Tables

**Figure 1 jcm-09-00573-f001:**
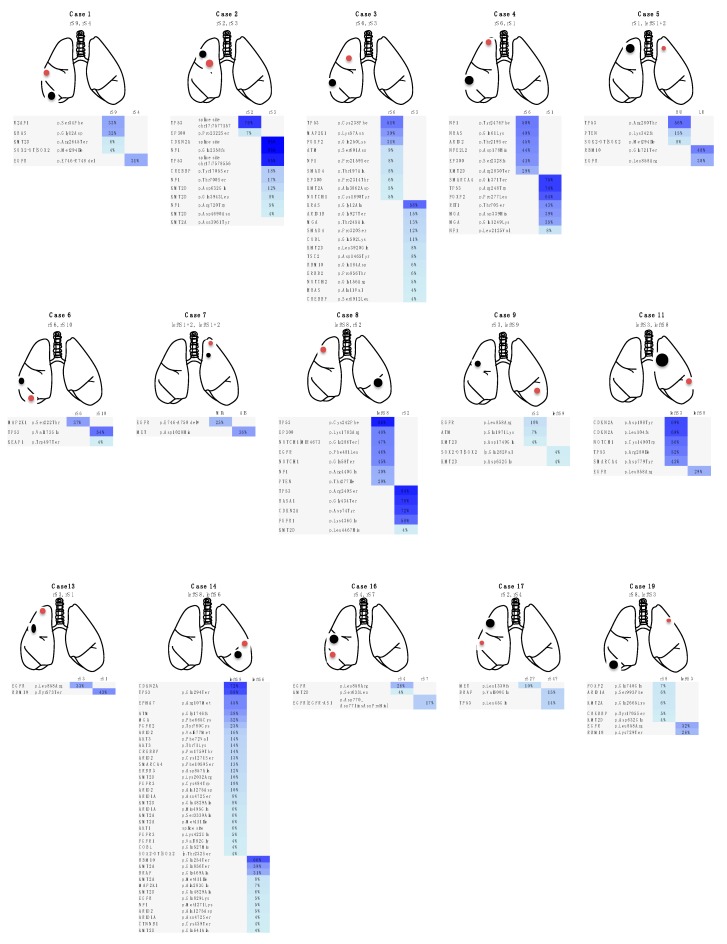
Heatmap of gene mutations in patients with double or triple primary lung cancers. These maps visualize the gene mutations in each cancer. Two or three lung cancers in each patient were characterized by different mutation profiles and all patients were diagnosed with double or triple primary lung cancers. Case 21, 22 and 28 were metachronous cancers, while the other cases in this figure were synchronous cancers. The remaining 5 cases of double primary lung cancers (cases 12, 18, 24, 26 and 34 in Table 1 and Appendix A) that are not shown in this figure are described in detail in the Case presentation section. Black, red and blue indicate tumor 1 (T1), T2 and T3, respectively. r, right; S, segment; AF, allele fraction; MIA, microinvasive adenocarcinoma; AIS, adenocarcinoma in situ.

**Figure 2 jcm-09-00573-f002:**
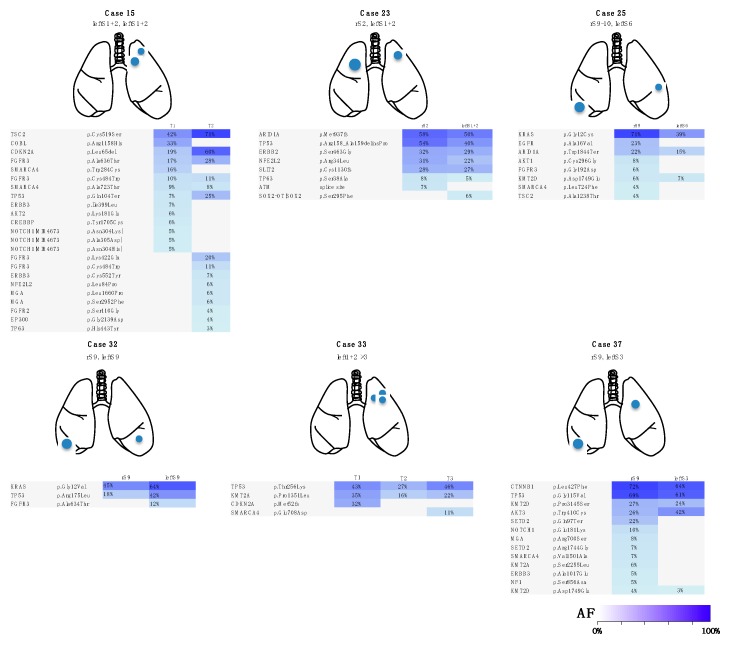
Heatmap of gene mutations in patients with metastatic lung cancers. The mutation profiles were consistent between the individual tumors in each case and the tumors were identified as intrapulmonary metastasis. Case 33 was synchronous cancers, while the other cases in this figure were metachronous cancers. The remaining 2 cases of metastatic lung cancers (cases 10 and 30 in Table 1 and Appendix A) that are not shown in this figure are described in detail in the Case presentation section. r, right; S, segment; AF, allele fraction.

**Figure 3 jcm-09-00573-f003:**
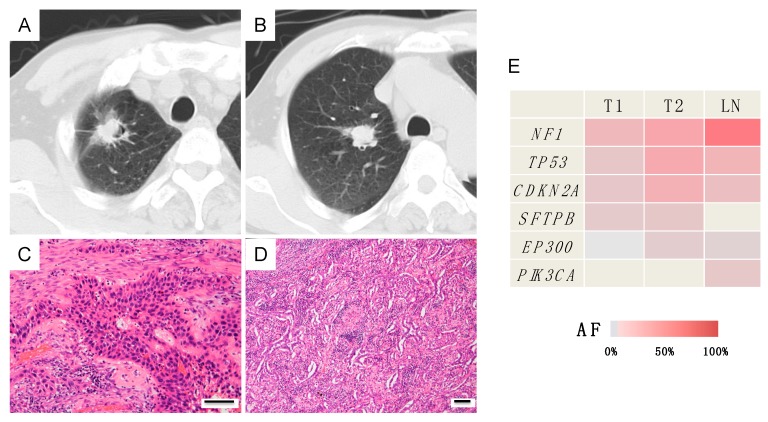
Radiological, histopathological and genomic findings in case A. (**A**,**B**) Right upper lobe nodules: one tumor was located in the peripheral region, whereas the other was located in the central region. (**C**) Histologically, the peripheral tumor (T1) was identified as an adenosquamous carcinoma. (**D**) The central tumor (T2) was histologically identified as an adenocarcinoma. Each scale bar indicates 100 μm. (**E**) The heatmap revealed that the same mutation profiles were shared by the two tumors and the lymph node metastasis. AF, allele fraction; LN, lymph node

**Figure 4 jcm-09-00573-f004:**
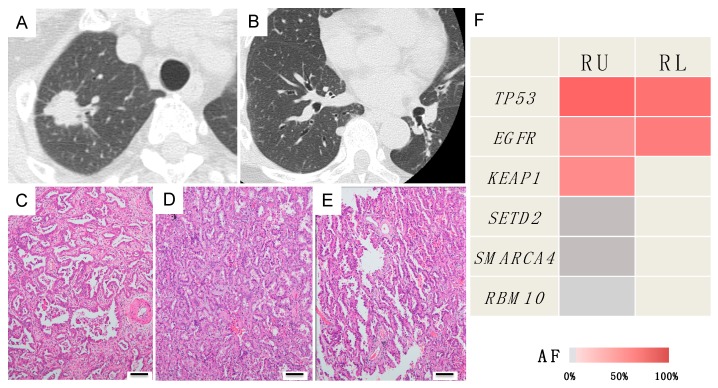
Radiological, histopathological and genomic findings in case B. (**A**) Lung cancer in the right upper lobe. (**B**) A small nodule in the right lower lobe. (**C**) Histology of the lung cancer in the right upper lobe. (**D**,**E**) Histology of the nodule in the right lower lobe. A lepidic pattern was observed in the periphery of the small nodule. Each scale bar indicates 100 μm. (**F**) Heatmap of the gene mutations of the two lung tumors. The significant mutations identified in the right upper lobe tumor were homologous with those detected in the right lower lobe tumor. RU, right upper lobe; RL, right lower lobe; AF, allele fraction

**Figure 5 jcm-09-00573-f005:**
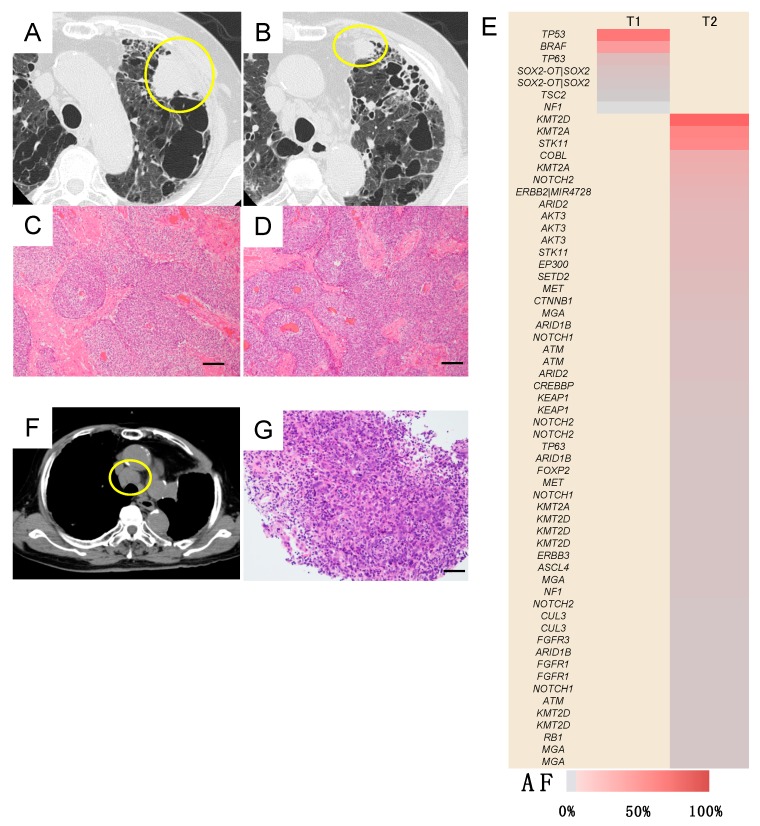
Radiological, histopathological and genomic findings in case C. (**A**,**B**) Two tumors, a large one (T1) and a small one (T2), were located in the left upper lobe in proximity to each other. (**C**,**D**) The tumors exhibited a similar histology of squamous cell carcinoma. Each scale bar indicates 100 μm. (**E**) Heatmap of the gene mutations of the two lung tumors. The mutation profiles of T1 and T2 were completely different. (**F**,**G**) Postoperatively, tracheobronchial lymph node enlargement was observed and the tumor was identified as a squamous cell carcinoma. Each scale bar indicates 100 μm. AF, allele fraction.

**Figure 6 jcm-09-00573-f006:**
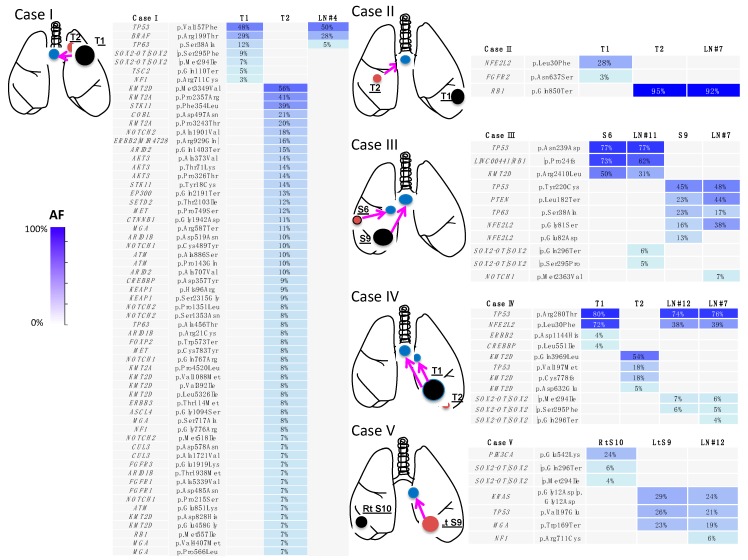
Schema of lymphatic metastasis and mutation profiles in multiple lung cancers. On the basis of the coincidence and differences in the mutation profiles, the clonality of each tumor and the pathway of lymphatic progression are clearly elucidated in each case. The arrows indicate the lymphatic routes of the cancer invasion. Tumor 1 is shown in black, tumor 2 in red and lymph node metastasis in blue. #4, tracheobronchial lymph node; #7, subcarinal lymph node; #11, interlobar lymph node; and #12, hilar lymph node. S, segment; LN, lymph node; AF, allele fraction

**Figure 7 jcm-09-00573-f007:**
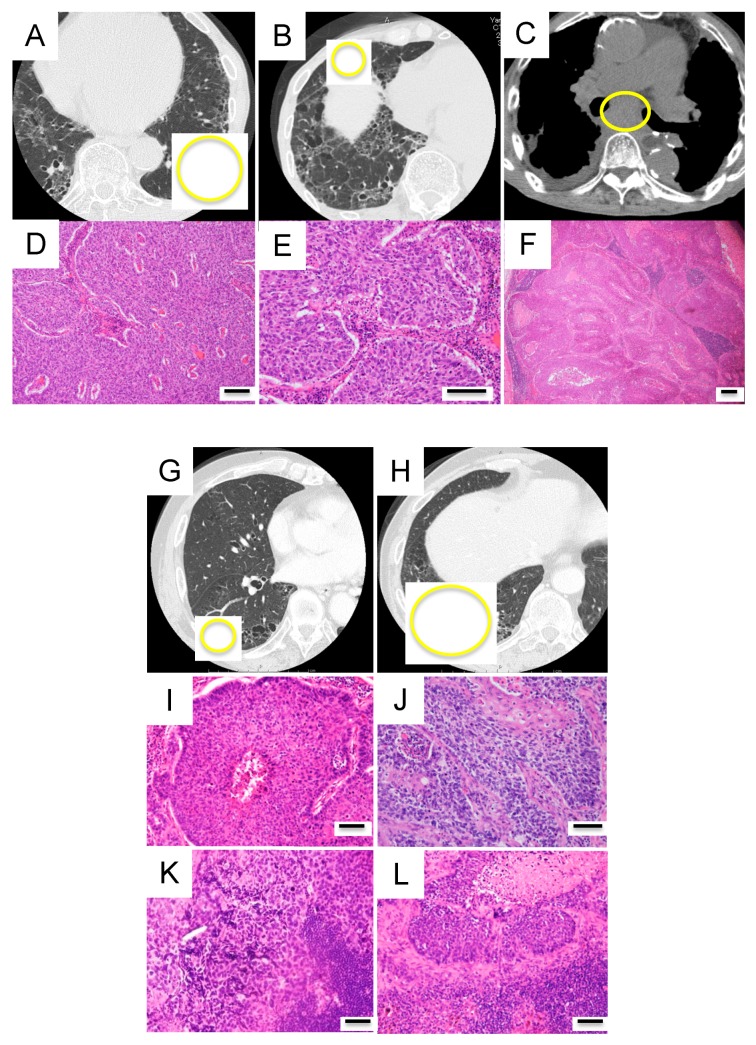
Radiological and histopathological findings in cases E and F. (**A**–**F**) Findings in case E. (**A**) Primary lesion in the left lower lobe. (**B**) Primary lesion in the middle lobe. (**C**) Subcarinal lymph node metastasis. (**D**–**F**) The three lesions displayed a similar histology of squamous cell carcinoma. (**G**–**L**) Findings in case F. (**G**) Primary lesion in right segment 6. (**H**) Primary lesion in right segment 9. (**I**) Histology of the primary lesion in segment 6. (**J**) Histology of the primary lesion in segment 9. (**K**) Histology of the subcarinal lymph node. (**L**) Histology of the interlobar lymph node. Histologically, the four lesions displayed a similar histology of squamous cell carcinoma. Each scale bar indicates 100 μm.

**Table 1 jcm-09-00573-t001:** Mutation analysis of the multiple lung cancers.

Case	Occurrence of Tumors	Interval between the 1st and 2nd Tumors	Clinical Dx	Pathological Dx	Genomic Dx
1	Synchronous	-	Double	Double	Double
2	Synchronous	-	Double	Double	Double
3	Synchronous	-	Double	Double	Double
4	Synchronous	-	Metastasis	Metastasis	Double
5	Synchronous	-	Double	Double	Double
6	Synchronous	-	Double	Double	Double
7	Synchronous	-	Double	Double	Double
8	Synchronous	-	Double	Double	Double
9	Synchronous	-	Double	Double	Double
10	Metachronous	14 months	Metastasis	Double	Metastasis
11	Synchronous	-	Double	Double	Double
12	Synchronous	-	Metastasis	Double	Double
13	Synchronous	-	Double	Double	Double
14	Synchronous	-	Double	Double	Double
15	Metachronous	15 months	Double	Metastasis	Metastasis
16	Synchronous	-	Double	Double	Double
17	Synchronous	-	Double	Double	Double
18	Synchronous	-	Metastasis	Metastasis	Double
19	Synchronous	-	Double	Double	Double
20	Synchronous	-	Double	Double	Double
21	Metachronous	17 months	Double	Double	Double
22	Metachronous	28 months	Double	Double	Double
23	Metachronous	23 months	Double	Metastasis	Metastasis
24	Metachronous	37 months	Double	Double	Double
25	Metachronous	37 months	Double	Double	Metastasis
26	Synchronous	-	Double	Double	Double
27	Synchronous	-	Double	Double	Double
28	Metachronous	41 months	Double	Double	Double
29	Synchronous	-	Double	Double	Double
30	Synchronous	-	Double	Double	Metastasis
31	Synchronous	-	Double	Double	Double
32	Metachronous	16 months	Double	Metastasis	Metastasis
33	Synchronous	-	Double	Metastasis	Metastasis
34	Metachronous	13 months	Double	Double	Double
35	Synchronous	-	Double	Double	Double
36	Synchronous	-	triple	triple	triple
37	Metachronous	46 months	Double	Double	Metastasis

The cases in which the diagnoses were inconsistent in the clinicopathological and genetic examinations are highlighted in gray.

**Table 2 jcm-09-00573-t002:** LN metastasis in patients with multiple lung cancers.

Case	Case No. in Table 1	Location and Histology of T1	Location and Histology of T2	Location and Histology of LN	LN Biopsy Method	Occurrence of LN Metastasis	Inconsistency between Clinical and Genomic Diagnoses
I	18	Left upper, Sq	Left upper, Sq	Right tracheobronchial, Sq	EBUS-TBNA	Postoperative	+
II	34	Left lower, Sq	Middle, Sq	Subcarinal, Sq	EBUS-TBNA	Postoperative	+
III	26	Right S9, Sq	Right S6, Sq	Interlobar and subcarinal, Sq	Surgery	Simultaneous	+
IV	12	Left S6, Sq	Left S10, small	Subcarinal, Sq	Surgery	Simultaneous	−
V	24	Right lower, acinar Ad	Left lower, solid Ad	Left lobar, solid Ad	Surgery	Simultaneous	−

S, segment; LN, lymph node; Sq, squamous cell carcinoma; Ad, adenocarcinoma; small, small cell carcinoma; EBUS-TBNA, endobronchial ultrasound-guided transbronchial needle aspiration.

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
