# Peer review of "Identification of Clonality through Genomic Profile Analysis in Multiple Lung Cancers"

_jcm, 2020, doi:10.3390/jcm9020573_

Round 1

Reviewer 1 Report

Thank you for your effort in the establishments for this research. The profiling methods are very well accomplished in other cancer centers in the world.

I would suggest investigating more detail of the discrepancy of genomic and clinical diagnosis which will be a great useful resource for those who are working on the diagnosis by genomic data. Possibly adding copy number variation analysis repurposing the genomic cohort in this study and/or compliments the gene expression (either microarray or RNAseq)

Author Response

I would suggest investigating more detail of the discrepancy of genomic and clinical diagnosis which will be a great useful resource for those who are working on the diagnosis by genomic data. Possibly adding copy number variation analysis repurposing the genomic cohort in this study and/or compliments the gene expression (either microarray or RNAseq)

Response:

Thank you very much for a high evaluation of our manuscript.

The key point of our study is that cancer-specific mutations which can be detected by the conventional targeted sequencing, i.e. point mutation and/or indels, serve sufficiently as clonal markers for establishing the genomic diagnosis. So far, all 37 cases were sufficiently characterized by these mutation profiles alone. The other genomic information, such as copy number variation and gene expression, is not deemed contributory to a more robust diagnosis, at least, in clinical settings. Therefore, we feel that the application of these genomic data (CNV or RNAseq) is beyond the scope of our current study. I personally believe this design (i.e. CNV and RNAseq in multiple lung cancers) may be our next research theme for those who are working on the diagnosis with genomic data, as the reviewer kindly suggested.

Thank you again for your thoughtful comments.

Reviewer 2 Report

This article by Higuchi et al describes the analysis of genomic alterations in lung cancers with multiple lesions, aiming to distinguish multiple primary lesions from metastatic disease, which cannot be resolved through histopathological or clinical characteristics.

The authors analyse 37 cases and provide specific details on 6 cases. Some cases are synchronous, other metachronous. Does this impact the interpretation of the results? In section 3.4, the authors discuss the synchronous and metachronous lesions briefly. Is there a higher frequency of double primary lesions in synchronous compared to metachronous cancer? To help with the interpretation of the results, could the authors indicate in table 1, and Figure 1 and 2, which tumours are synchronous or metachronous. For each case of metachronous tumours, can the latency between the detection of the different tumours be indicated.

Existence of intra-tumour heterogeneity has been extensively described by C Swanton’s group. Within a primary tumour, differences exist in mutation status (Jamal-Hanjani et al., NEJM 2017 and others). This intratumour heterogeneity could impact the interpretation of the results, where a mutation found in a metastatic lesion may not be there in the sample taken of the primary lesion, but may be there in another region of the same primary tumour. This should be discussed in the present manuscript.

For ease of read and comparison between tables and figures, could the same case ID be used consistently across the manuscript. For example, case 18 in table 1, is case C in section 3.3, then case D in section 3.6, and case 1 in table 2, and case I in figure 6.

Author Response

Comment 1: The authors analyse 37 cases and provide specific details on 6 cases. Some cases are synchronous, other metachronous. Does this impact the interpretation of the results? In section 3.4, the authors discuss the synchronous and metachronous lesions briefly. Is there a higher frequency of double primary lesions in synchronous compared to metachronous cancer? To help with the interpretation of the results, could the authors indicate in table 1, and Figure 1 and 2, which tumours are synchronous or metachronous. For each case of metachronous tumours, can the latency between the detection of the different tumours be indicated.

Response: In answering to the question as to whether there is a higher frequency of double primary lesions in synchronous compared to metachronous cancer, we would like to say yes to that question. To make this point clear, we added some descriptions, as follows.

“The distribution of primary and metastatic tumors between synchronous and metachronous tumors was significantly different; thus synchronous multiple lung tumors were deemed likely to be primary lesions.”

In order to indicate which tumors are synchronous or metachronous, we added some descriptions in the legend of Figures 1 and 2. In addition, Table 1 was modified according to the reviewer’s suggestion.

Comment 2: Existence of intra-tumour heterogeneity has been extensively described by C Swanton’s group. Within a primary tumour, differences exist in mutation status (Jamal-Hanjani et al., NEJM 2017 and others). This intratumour heterogeneity could impact the interpretation of the results, where a mutation found in a metastatic lesion may not be there in the sample taken of the primary lesion, but may be there in another region of the same primary tumour. This should be discussed in the present manuscript.

Response: In the Discussion section, we added some explanations about the “trunk and branch” mutation models, which Dr. Swanton previously advocated in relation to intratumor heterogeneity. Also, the paper (Jamal-Hanjani et al., NEJM 2017 and others) was cited as a new reference in the revised manuscript.

Comment 3: For ease of read and comparison between tables and figures, could the same case ID be used consistently across the manuscript. For example, case 18 in table 1, is case C in section 3.3, then case D in section 3.6, and case 1 in table 2, and case I in figure 6.

Response: According to the reviewer’s suggestion, we simplified the case ID, not to confuse the readers.

Thank you again for your thoughtful comments.